# Comprehensive Transcriptome Analysis Reveals the Distinct Gene Expression Patterns of Tumor Microenvironment in HPV-Associated and HPV-Non Associated Tonsillar Squamous Cell Carcinoma

**DOI:** 10.3390/cancers15235548

**Published:** 2023-11-23

**Authors:** Reham M. Alahmadi, Najat Marraiki, Mohammed Alswayyed, Hatim A. Khoja, Abdullah E. Al-Anazi, Rawan M. Alahmadi, Meshael M. Alkusayer, Bandar Alosaimi, Maaweya Awadalla

**Affiliations:** 1Department of Botany and Microbiology, College of Science, King Saud University, P.O. Box 22452, Riyadh 11495, Saudi Arabia; realahmadi@ksu.edu.sa (R.M.A.);; 2Department of Pathology and Laboratory Medicine, College of Medicine, King Saud University, P.O. Box 22452, Riyadh 11495, Saudi Arabia; malswayyed@ksu.edu.sa; 3Department of Pathology and Laboratory Medicine, King Faisal Specialist Hospital and Research Center, Riyadh 11211, Saudi Arabia; hkhoja@kfshrc.edu.sa; 4Comprehensive Cancer Center, King Fahad Medical City, Riyadh Second Health Cluster, Riyadh 11525, Saudi Arabia; 5Head and Neck Surgery Division, Department of Otolaryngology/Head and Neck Surgery, Prince Sultan Military Medical City, P.O. Box 7897, Riyadh 11159, Saudi Arabia; 6Department of Otolaryngology-Head and Neck Surgery, King Saud Medical City, Riyadh 11525, Saudi Arabia; malkusayer@ksmc.med.sa; 7Department of Research Labs, Research Center, King Fahad Medical City, Riyadh Second Heath Cluster, Riyadh 11525, Saudi Arabia

**Keywords:** HPV, tonsillar squamous cell carcinoma, gene expression, tumor microenvironment, inflammation, oncogenes, tumor suppressor gene, immunity crosstalk

## Abstract

**Simple Summary:**

The tumor microenvironment (TME) of HNSCC is heterogeneous and complex, and it plays a crucial role in achieving effective cancer therapy. To develop effective cancer therapies, it is important to understand the crosstalk between cancer inflammation and immunity, as well as oncogenes and tumor suppressor genes. In this study, we aimed to gain a more comprehensive understanding of the transcriptomes of the TME in HPV-associated and HPV-non-associated TSCC by studying the gene expression profiles of 168 genes linked to various cellular mediators and factors involved in inflammation, immune crosstalk, transcription, signal transduction, oncogenesis, tumor suppression, angiogenesis, and apoptosis. We found that the TME of HPV-associated and HPV-non-associated TSCC exhibited a remarkable heterogeneity of gene expression associated with cellular mediators and factors involved in inflammation, immune crosstalk, transcription factors, immune signaling pathways, signal transduction, oncogenesis, tumor suppression, angiogenesis, and apoptosis.

**Abstract:**

Head and neck squamous cell carcinomas (HNSCCs) are a common type of cancer, ranking as the sixth most prevalent cancer worldwide and having a high morbidity and mortality rate. Among oropharyngeal squamous cell carcinoma (OPSCC) cancers, tonsillar squamous cell carcinoma (TSCC) is the most prevalent and has a particularly aggressive clinical course with poor disease outcomes. The tumor microenvironment (TME) of HNSCC is complex and heterogeneous, playing a crucial role in effective cancer therapy. Understanding the interaction between cancer inflammation, immunity, oncogenes, and tumor suppressor genes is essential for developing effective cancer treatments. This study aimed to gain a comprehensive understanding of the transcriptomes of the TME in TSCC, both associated with human papillomavirus (HPV) and not associated with HPV. The gene expression profiles of 168 genes linked to various cellular mediators and factors involved in inflammation, immunity crosstalk, transcription, signal transduction, oncogenesis, tumor suppression, angiogenesis, and apoptosis were analyzed. We identified 40 differentially expressed genes related to the communication between tumor cells and the cellular mediators of inflammation and immunity crosstalk. In HPV-positive TSCC patients, 33 genes were over-expressed with a fold change greater than 1.5, and 26 of these genes were unique to this group. In contrast, HPV-negative TSCC patients had 11 up-regulated genes. The results further showed that 48 gene transcripts related to oncogenesis, tumor suppression, angiogenesis, and apoptosis were up-regulated in both HPV-positive and HPV-negative TSCC patients. Among the HPV-positive TSCC patients, 37 genes were over-expressed, while the HPV-negative TSCC patients had 11 up-regulated genes. The tumor microenvironment (TME) of HPV-associated and HPV-non-associated TSCC exhibited distinct characteristics, including the dysregulation of various genes involved in cellular mediators, inflammation, immunity crosstalk, transcription factors, immune signaling pathways, signal transduction, oncogenesis, tumor suppression, angiogenesis, and apoptosis. Additionally, we detected six Hr-HPV genotypes in 81% of the TSCC patients, with HPV-16 and HPV-35 being the most common types, followed by HPV-45 and HPV-18. HPV-39 and 31 were also identified. The presence of Hr-HPV genotypes in TSCC patients varied from single to multiple infections. In conclusion, we observed distinct heterogeneity in the transcriptome of the microenvironment in HPV-associated and non-associated TSCC. Further in vitro and in vivo studies are needed to investigate the functional implications of the identified over-expressed genes. Also, deeper molecular pathways and immunological studies on the TME are required to determine the potential of targeting genes for cancer therapy.

## 1. Introduction

Head and neck squamous cell carcinomas (HNSCCs) have a high morbidity and mortality rate and rank as the sixth most common cancer globally, with approximately 900,000 new cases and more than 450,000 deaths in the last two years [1,2]. By 2030, the incidence of HNSCCs is expected to reach around 1.08 million new cases annually (Global Cancer Observatory (GLOBOCAN)). In 2020, among the total Saudi population (34,813,867), there were 27,885 new cancer cases with a mortality rate greater than 13,069 deaths [2]. The incidence of HNSCCs in Saudi Arabia has been sharply increasing, with up to 3730 cases each year [3]. HNSCCs account for around 13% of all cancers in Saudi Arabia [4]. Despite significant advancements in diagnosis and treatment, the 5-year survival rate of HNSCCs has not significantly improved [5]. HNSCC patients have a poor prognosis, with a five-year survival rate of 40–50% [6]. Among the HNSCCs, oropharyngeal squamous cell carcinoma (OPSCC) is considered one of the most common cancers [7]. Approximately 50% of oropharynx tumors are associated with high-risk human papillomavirus (hr-HPV), primarily HPV-16, HPV 31, HPV 18, HPV 56, HPV 52, HPV 33, and HPV 35, as well as other hr-HPV genotypes [8,9], placing tonsilla squamous cell carcinoma (TSCC) as the most invasive of the oropharyngeal cancers, accounting for more than 23% of all oropharyngeal malignancies [7,10,11]. The HPV genome codes for six oncogenes are named E1, E2, E4, E5, E6, and E7 [12,13]. These genes play a crucial role during viral replication and tumorigenesis [13]. E6 degrades the tumor suppressor p53 and ensures the survival of infected cells, while E7 promotes proliferation and inactivates the retinoblastoma protein (pRB) [14]. E6 and E7 could also cause genomic instability. The integration of HPV DNA into the human genome and E6 and E7 persistent expression is considered a key event in the progression of HPV-associated cancers [15].

TSCC has an aggressive clinical course and poor disease outcomes [16]. Around 60–80% of TSCC patients are found to have systemic metastasis at initial diagnosis, which indicates early dissemination [17]. The HNSCC treatment options are based on surgery with adjuvant standard radiotherapy or chemotherapy and immunotherapy. Despite intensive research, the treatment outcomes are poor due to the complexity of the anatomical structure of the head and neck region [18]. The levels of anticancer drug responsiveness are distinct among TSCC patients [19].

The tumor microenvironment (TME) of HNSCC is a crucial factor in achieving effective cancer therapy [6,20,21]. The TME of HNSCC is heterogeneous and complex, consisting of a variety of cell types, including cancer-associated fibroblasts, diverse immune cells, and non-cell extracellular matrix components [22,23]. Constant crosstalk and interaction between tumor cells and the surrounding microenvironment influence each other, either positively or negatively [24], by either promoting or inhibiting tumor growth, progression, and metastasis [25,26]. The TME can also affect the outcome of cancer immunotherapy by modulating immune cell infiltration and/or activation [27]. Immune checkpoint inhibitors can be used to modulate immune cells in the TME by blocking immune checkpoint molecules. The crosstalk between cancer, inflammation, and immunity encompasses complex molecular events and interactions between immune cells and cancer cells, leading to the production of inflammatory mediators, the recruitment of immune cells, and the modulation of gene expression [28,29]. The crosstalk between cancer and immune cells is mediated by various molecules, including inflammatory cytokines, chemokines, and growth factors. This complex interplay affects tumor development, progression, immune response to cancer, and therapy [28]. Cancer cells can modulate the inflammatory response, leading to a chronic inflammatory yet immunosuppressive microenvironment that promotes tumor growth and metastasis. The crosstalk between cancer and immune cells is mediated by various molecules, including inflammatory cytokines, chemokines, and growth factors [29,30]. Oncogenes and tumor suppressor genes play a role in cancer. Oncogenes are genes that have the potential to initiate cancer when they are mutated or over-expressed [31]. Tumor suppressor genes, on the other hand, are genes that normally control cell cycle division or programmed cell death. Mutations or loss of function of tumor suppressor genes can lead to cancer [31]. Unlike oncogenes, tumor suppressor genes negatively regulate cell cycle control and exhibit loss-of-function alterations in cancer [32]. Understanding the crosstalk between cancer, inflammation, immunity, oncogenes, and tumor suppressor genes is of great significance for developing effective cancer therapies that target the TME and enhance antitumor immunity. To gain a more comprehensive understanding of the TME transcriptome in HPV-associated TSCC and HPV-negative TSCC, we studied the gene expression profiles of 168 genes that are linked to communication between tumor cells and the cellular mediators of inflammation, immunity, transcription and signal transduction factors, oncogenesis, tumor suppressors, transcription factors, angiogenesis, and apoptosis.

## 2. Materials and Methods

### 2.1. Sample Collection and Participant Characteristics

Sixteen formalin-fixed paraffin-embedded tissue samples of TSCC were collected from four different hospitals, King Saud Medical City, King Fahad Medical City, King Faisal Specialist Hospital & Research Center and Dr. Sulaiman Al-Habib Medical Group, between 2014 and 2021. TSCC was confirmed by two pathologists. The preliminary inclusion criteria used in this study were (1) both males and females aged above 18 years; (2) the presence of invasive TSCC; (3) sufficient material for HPV detection and genotyping; and (4) an available block of formalin-fixed, paraffin-embedded tissue for the transcriptome of the invasive carcinoma. Patients who received chemotherapy or radiotherapy for any cancer, pregnant women, immunocompromised patients, patients with immune-mediated diseases, those treated with immunotherapy, or those with incomplete or missing records or data, as well as those under the age of 18, were excluded from the study. A series of 10 μm-thick tissue sections were cut from each paraffin block. To ensure the accuracy of the results, standard measures were taken to avoid cross-contamination during tissue sectioning and processing.

### 2.2. Ethical Considerations

The study protocol was reviewed and approved by the Institutional Review Board (IRB) committee at King Saud University, College of Medicine (IRB number E-22-6932).

### 2.3. Simultaneous Isolation of Viral DNA and Total RNA

Viral DNA total RNA was extracted using the AllPrep DNA/RNA FFPE Kit from Qiagen (Hilden, Germany) following the manufacturer’s protocol. The concentration (ng/μL) and purity of the extracted nucleic acid were assessed using a NanoDrop2000 (Thermo Fisher Scientific, Waltham, MA, USA). cDNA synthesis and genomic DNA elimination were performed using an RT2 first-strand synthesis kit (Qiagen, Germantown, MD, USA) with 1.5 µL of RNA, following the manufacturer’s instructions. The integrity of the extracted nucleic acid and the absence of inhibitors were assessed using two internal housekeeping genes (GAPDH and HMBS). We have restricted measurements for handling the total RNA, such as working in a biological safety cabinet and in RNase-free environments.

### 2.4. HPV Detection, Genotyping, and HPV E Oncogenes Integration Detection

For HPV detection and genotyping, we used qPCR using a SYBR Green/ROX PCR Master Mix according to the manufacturer’s protocol. The extracted total RNA was converted into cDNA using the RT2 first-strand synthesis kit (Qiagen, Germantown, MD, USA) based on the manufacturer’s recommendations to detect HPV E6/E7 mRNA expression. The reaction contained 10 µL of 2× RT2 SYBR^®^ Green Master mix, 1 µL of forward primer, 1 µL of reverse primer, 2 µL of cDNA, and 6 µL of RNase-free water. The cycling program was as follows: initial denaturation at 95 °C for 10 min, followed by 40 cycles; denaturation at 95 °C for 15 s; annealing at 57 °C and extension at 72 °C for 45 s using the ABI-7500 Fsat (Applied Biosystems, Waltham, MA, USA). The list of specific primers can be found in Appendix A.

### 2.5. Cellular RNA Processing and Cancer Inflammation, Immunity Crosstalk, Oncogenes, and Tumor Suppressor Genes Transcriptomes

For mRNA expression, cDNA synthesis and genomic DNA elimination were performed using an RT2 first-strand synthesis kit (Qiagen, Germantown, MD, USA) following the manufacturer’s instructions. The synthesized cDNA was used for an RT-PCR array of cancer inflammation and immunity crosstalk (PAHS-181Z), oncogenes, and tumor suppressor genes (PAHS-502ZC-12) (SABiosciences, located in Frederick, MD, USA). These two RT^2^ profiler PCR arrays were used for the expression of 168 genes that are linked to communication between tumor cells and the cellular mediators of inflammation and immunity, oncogenesis promotion, tumor suppressor genes, transcription factors, angiogenesis, and apoptosis. The detailed list of all genes in both RT2 Profiler PCR Array Kits can be checked on the manufacturer’s website (https://geneglobe.qiagen.com/us/product-groups/rt2-profiler-pcr-arrays, accessed on 29 August 2023). Real-time PCR was performed using the RT2 Profiler PCR Array Kit (Qiagen, Germantown, MD, USA) in a mixture with the RT2 SYBR Green/ROX qPCR Master Mix (Qiagen), as previously described [33]. This is a ready-to-use PCR master mix that contains SYBR Green dye for fluorescence detection and ROX as a passive reference dye. Briefly, one 96-well plate, with a total volume of 2700 µL, contains 1350 µL of 2× RT2 SYBR^®^ Green Master mix, 102 µL of cDNA, and 1248 µL of RNase-free water. Subsequently, 25 µL of the qPCR component mix was added to each well of the PCR array, and then the 96-well array was placed in the ABI-7500 Fast (Applied Biosystems, Waltham, MA, USA) using 40 cycles at 95 °C, with 10 min per cycle, followed by 95 °C for 15 s and 60 °C for 1 min for annealing, extension, and fluorescence data collection. The online RT2-PCR Profiler data PCR array analysis software was used for relative gene expression results (RT^2^ Profiler PCR Array Data Analysis, www.qiagen.com). Gene expression was quantified as ΔΔCt, and calculations were normalized to the five housekeeping genes.

### 2.6. Data Analysis

GraphPad 10 software (GraphPad Software, San Diego, CA, USA) was used for data analysis. A group comparison was performed using a *t*-test. The correlations between all variables were assessed using Pearson’s correlation test. A *p*-value of < 0.05 was considered statistically significant. For gene expression and real-time PCR array data analysis, online RT2 Profiler Data Analysis software was used. The fold regulation values for each gene were determined using the relative quantification 2-ΔΔCt method. ΔCt values were normalized using the mean values of five housekeeping genes.

### 2.7. Protein–Protein Interaction Network and Hub Protein Analysis

A functional analysis of statistically significant over-expressed genes was used for the construction of the biological network using the Retrieval of Interacting Genes/Proteins database (STRING version 11.0) and Cytoscape software 3.10.1.

## 3. Results

### 3.1. Basic TSCC Patient’s Characteristics

The total FFPE samples from TSCC patients (seven males and nine females) and three individuals without TSCC were included in this study. The age range was 24–70 years. The median age was 57 ± 12. The complete demographics and clinical characteristics are shown in Table 1.

### 3.2. Prevalence of HPV in Tonsil Squamous Cell Carcinoma (TSCC)

From the PEPP tissue-extracted DNA, HPV DNA was detected in TSCC patients. Thirteen (81.3%) were HPV-positive, while 18.8% had HPV-negative TSCC. We detected six Hr-HPV genotypes in 81% of the TSCC patients. The most common types were HPV-16 and HPV-35 (62%), followed by HPV-45 (25%) and HPV-18. HPV-39 and 31 (6%) were the least common types in this study. The presence of HPV in TSCC patients varied from single to multiple infections. Single HPV-16 and HPV-35 infections were detected in three patients (60%) and two patients, respectively. Whereas double HPV infections (25%) (HPV 16 + HPV 35) were detected in two TSCC patients (Table 2). On the other hand, five (62%) TSCC patients had different triple HPV infections (HPV 16, 35, and 45), e.g., in two patients: HPV 16, 18, and 35 and HPV 18, 35, and 45 (Table 2). Surprisingly, we detected quadruple HPV infections in one TSCC patient.

### 3.3. Cancer Inflammation and Immunity Crosstalk Gene Expression Analysis of TME in HPV-Associated and Non-Associated TSCC

Cancer, inflammation, and immunity are closely interconnected, and their crosstalk plays a critical role in tumor development and progression. Inflammation can promote the initiation of cancer by inducing DNA damage and genetic mutations. It can also create a microenvironment that supports tumor growth by enhancing angiogenesis and suppressing immune surveillance. Therefore, we studied the mRNA expression profile of immune genes involved in crosstalk between the cancer, inflammation, and immunity pathways using the Human Cancer Inflammation and Immunity Crosstalk RT2 Profiler PCR Array in HPV-positive TSCC and HPV-negative patients. A total of 84 key genes involved in mediating the communication between tumor cells and the cellular mediators of inflammation and immunity crosstalk were included in this qPCR array. Based on the internal positive controls, genomic DNA elimination control, and reverse transcription controls, this profile showed higher reproducibility and efficiency. We identified 40 genes related to communication between tumor cells and the cellular mediators of inflammation and immunity crosstalk that were differentially dysregulated in HPV-positive TSCC and HPV-negative patients (Figure 1 and Appendix A). Thirty-three of these genes were highly expressed in HPV-positive TSCC patients (Figure 1) (fold change > 1.5). Of these, the top 14 genes (VEGFA, MYC, CTLA4, CD274 (PDL-1), FOXP3, GBP1, CCL4, CCL18, CXCL8, EGFR, IL23A, HAL-C, IL4, and CCL5) were highly expressed in HPV-positive TSCC patients (Figure 1). However, HPV-negative TSCC patients showed an up-regulation of 10 genes; among them, 5 genes (IL6, CXCL9, CXCL12, CCL4, and IL13) were considered the most expressed genes (Figure 1). The analysis showed that six genes (CCL4, CSF2, CSF3, CXCL12, CXCL9, and IL6) were co-expressed and common between HPV-positive TSCC and HPV-negative patients. These genes showed a range of 1·6–17-fold change. Furthermore, when we compared the two groups of TSCC patients, the results revealed that four genes (CCR2, IL12B, IL13, and TNF) were up-regulated in the HPV-negative TSCC patients but not in the HPV-positive TSCC patients (Figure 1). The CCL4, CXCL12, and CSF3 gene expression levels were strongly increased in the HPV-associated TSCC patients compared with the HPV-non-associated TSCC patients (Figure 1). It is important to mention that two genes, namely IL-6 and CSF2, were not changed between the two groups. Regarding the down-regulated genes, we found that the HPV-negative TSCC patients were characterized by highly down-expressed immune-related genes (CXCR2, CXCL9, IL13, CSF3, and CCR7) compared to HPV-associated TSCC patients. There was a high heterogeneity in TME gene expression in HPV-associated and HPV-non-associated TSCC patients. There were two patients (p2 and p4) positive for seven genes (22%), one patient (p6) positive for one gene (11%), one patient (p3) positive for four genes (11%), one patient (p8) positive for six genes (11%), one patient (p9) positive for fourteen genes (11%), and one patient (p7) positive for twenty genes (11%), while in two patients (p1 and p5), we did not detect over-expressed genes at all (22%) (Figure 2). Overall, we observed distinct heterogeneity in the HPV-associated and HPV-non-associated TSCC microenvironment gene expression of cancer inflammation and immunity crosstalk. The detailed fold-change in the gene expression levels for each patient is presented in Figure 2a,b.

### 3.4. Gene Expression Analysis of Oncogenes and Tumor Suppressor Genes of TME in HPV-Associated and Non-Associated TSCC

The dysregulation of oncogenes and/or tumor suppressor genes plays a crucial role in the initiation and progression of TSCC. We aimed to explore the levels of mRNA expression of the 84 oncogenes and tumor suppressor genes in the TSCC microenvironment via the Human Oncogenes and Tumor Suppressor Genes RT^2^ Profiler™ PCR Array. We also wanted to investigate the interplay between them and inflammation-related genes. Our results showed that a total of 48 gene transcripts were up-regulated (fold change > 1.5) in the HPV-associated and the HPV-non-associated TSCC patients (Figure 3). In the HPV-associated TSCC patients, 37 genes were over-expressed; of those, 28 genes were unique in this group. However, the HPV-non-associated TSCC patients had only 11 up-regulated genes, and among them, only two genes (PIK3C2A and TSC1) were unique for this group (Figure 3). We compared both groups of TSCC patients and found that nine genes were co-expressed (Figure 3). This heterogenicity between HPV-associated and HPV-associated TSCC could aid in exploring and understanding the molecular complexity of HPV–host interactions, which may lead to a new cancer therapy for TSCC. To classify and identify the up-regulated genes, we clustered those genes into seven clusters (C1, C2, C3, C4, C5, C6, and C7). The most up-regulated clusters were oncogenes, C1 (19 genes), followed by tumor suppressors, C2 (14 genes), transcription factors, C3 (8 genes), and dual C4 (4 oncogenes and tumor suppressor genes), angiogenesis, C5 (1 gene), cell adhesion molecules, C6 (1 gene), and cell cycle, C7 (1 gene). It is important to note that the TME of HPV-non-associated TSCC patients is characterized by only three clusters: oncogenes, tumor suppressors, and transcription factors. There was a high heterogeneity in the TME gene expression in the HPV-associated and HPV-non-associated TSCC patients (Figure 3). There were two patients (p6 and p8) positive for nine genes (22%); one patient (p4) positive for three genes (11%); one patient (p3) positive for six genes (11%); one patient (p2) positive for ten genes (11%); one patient (p9) positive for twenty genes (11%); one patient (p5) positive for twenty-one genes (11%), and one patient (p7) positive for twenty-four genes (11%). In one patient (p1), no over-expressed genes were detected (22%). The detailed fold-change in the gene expression levels for each patient is presented in Figure 4a,b.

### 3.5. No Association between the Most Up-Regulated Gene Expression and Age

Age is one of the factors that affect the immunity status of patients. Therefore, we investigated a potential correlation between over-expressed genes among most the TSCC patients and age. A Person’s correlation analysis showed that there was no significant positive correlation between the CAPS8, FOS, STK11, CCL5, and CSF3 genes and age, while a negative association between CDKN2B, NCL1, and MDM2 and age was found.

### 3.6. Construction of a Protein–Protein Interaction Network

The PPI network was constructed utilizing the online STRING website (cn.string-db.org). The most up-regulated genes (CCL4, CSF2, CSF3, CXCL12, IL6, NF1, NFKBIA, RAF1, RARA, STK11, WWOX, BAX, ESR1, and MCL1) were input into the gene list, and homo sapiens were selected. The medium confidence level was set to 0.400. Then, we hid the disconnected nodes of the network and adjusted the position of each node. The PPI contained 43 nodes, 372 edges, and an average node degree of 17.3, with a *p* value < 1.0 × 10^−16^. (Figure 5). The PPI network was clustered into three clusters via the k-means clustering option (Figure 6). This PPI contains different types of gene connections, such as the gene neighborhood, gene fusions, cooccurrence, coexpression, text mining, protein homology, and biochemical/genetic data (experiments), as well as direct (physical) or indirect (functional) interactions.

## 4. Discussion

The TME plays a crucial role in the initiation and progression of several types of cancer, making it a significant area of research in recent years. A deeper and more complete understanding of the specific components of the TME, particularly the immune oncogenes and tumor suppressor gene components, remains a significant challenge. To gain a more comprehensive understanding of the gene expression in the TME of HPV-associated and non-associated TSCC, we focused on the gene expression profiling of 168 genes that are linked to the communication between tumor cells and the cellular mediators of inflammation and immunity, oncogenesis promotion, tumor suppressor genes, transcription factors, angiogenesis, and apoptosis. The TME of HNSCC is a crucial factor in achieving effective cancer therapy [20,21]. The TME is heterogeneous and complex, consisting of a variety of cell types, including cancer-associated fibroblasts, diverse immune cells, and non-cell extracellular matrix components [22,23,34]. Constant crosstalk and interaction between tumor cells and the surrounding microenvironment influence each other, either positively or negatively [24], by either promoting or inhibiting tumor growth, progression, and metastasis [25,26]. The TME can also affect the outcome of cancer immunotherapy by modulating immune cell infiltration and/or activation [27]. Characterizing the TME of HNSCC has identified TME as a heterogeneous environment [20]. Different factors could lead to changes in gene expression patterns in the TME of HNSCC [21]. Some studies have studied the TME of HNSCC and identified important gene signatures as prognostic and immunotherapeutic targets [23]. In addition, the dysregulation of various genes in HNSCC patients has been associated with reduced overall survival [20].

The HPV-associated and non-associated TMEs are characterized by the dysregulation of immune-related genes, tumor suppressor genes, and oncogenes. This dysregulation may be attributed to the presence of HPV and its viral proteins, which can disrupt normal cellular processes and immune responses. Consequently, genes involved in immune surveillance and tumor suppression become dysregulated. Several studies have shown that the immune microenvironment plays a crucial role in the prognosis and targeted immune therapy of HNSCC patients [35,36,37,38]. Furthermore, the prediction of HNSCC patients’ prognosis was shown to be associated with different immune-related genes [39,40]. In another study, an immune-related gene-based prognostic signature was constructed and validated in HNSC samples based on clinical and transcriptomic data [37]. These studies highlight the prognostic importance of the immunological TME in HNSCC and demonstrate that the immune-related gene of the TME can significantly affect a patient’s prognosis.

In this study, we found that several immune-related genes were up-regulated (CCL4 and CCL5) in most of our cohorts, whereas BCL2, CCR7, CSF3, CXCL9, CXCR2, and IL13 were down-regulated. The CCL4 and CCL5 chemokines play a vital role in various cancer types in terms of metastasis, invasion, and poor clinical disease outcomes. Several studies have revealed that the up-regulation of CCL4 and CCL5 in gastric carcinoma, tongue squamous cell carcinoma (TSCC), and colon cancer resulted in the initiation of invasion steps [41,42,43]. Moreover, accumulating evidence showed the significant role of CCL4 in tumorigenesis development and cancer progression by altering the immune cells in the TME [44]. The over-expression of CCL4 in oral squamous cell carcinoma and lung cancer increased lymphangiogenesis and vascular endothelial growth factor c expression [45] and PD-L1 [45,46]. Targeting the CCL4 and CCL5 pathways may offer new strategies for controlling cancer metastasis and improving treatment outcomes in TSCC. By modulating the expression or activity of CCL4 and CCL5, it may be possible to modulate the immune cell composition and activity within the TME, thereby impacting the tumor immune response. Understanding the functional implications of gene alterations in the CCL4 and CCL5 pathways can guide the development of targeted therapies. Further studies are needed to investigate the functional implications of gene alterations in the CCL4 and CCL5 pathways in TSCC.

Tumor growth and metastasis in cancer have been linked to angiogenesis [47]. Several studies have shown that many important anti-angiogenic drugs can control cancer activity [47,48]. A higher expression of angiogenic genes, such as vascular endothelial growth factor (VEGF), has been associated with poor progression and treatment resistance in head and neck cancer [49,50]. In this study, we found higher expression levels of VEGFA and AKT1. We believe that targeting an angiogenic gene may be a potential strategy for TME modulation, which may lead to tumor treatment sensitivity. BCL2 gene expression has been extensively studied in various types of cancer, including breast cancer, colorectal cancer, bladder cancer, lymphoma, and acute myeloid leukemia (AML). The relationship between BCL2 gene expression and cancer is complex and can vary depending on the specific cancer type and context. Studies have shown that breast cancer patients with positive BCL2 expression tend to have better overall survival and disease-free survival [51]. In colorectal cancer, the prognostic significance of BCL2 expression is unclear. Some studies have suggested that BCL2 expression has no prognostic significance in colorectal cancer [52]. BCL2 expression has been associated with both positive and poor prognostic outcomes in different cancers [53,54]. Our results showed that the mRNA expression levels of BCL2 were down-regulated in the majority of our cohort. The down-regulation of BCL2 mRNA expression in our study may suggest a potential role for BCL2 as a tumor suppressor or an indicator of poor prognosis. The association between BCL2 gene expression and cancer is complex and can vary depending on the specific cancer type and context. The dysregulation of BCL2 expression has been reported in various cancers, with both up-regulation and down-regulation observed depending on the cancer type [51,52,53,54]. It is crucial to consider multiple factors, such as the heterogeneity and specific characteristics of the cancer type, the sample size, and potential confounding variables, when interpreting BCL2 expression. Therefore, future studies should conduct comprehensive analyses and integrate their findings with existing knowledge in the field to derive meaningful conclusions.

CXCL9 is one of the immune activation chemokines; its expression plays a crucial role in anti-tumor immunity and is associated with improvements in different types of cancer patients’ survival and overall good clinical outcomes [55,56]. In this study, we found that the expression levels of CXCL9 were down-regulated in the majority of our cohort. This is consistent with previous studies that reported that a lower expression of CXCL9 among several genes could be an early prognostic factor in HNSCC and non-small-cell lung cancer [57]. Some studies have reported that CXCL9 up-regulation in HNC squamous cell carcinoma contributes to worse disease outcomes [58]. Regarding the CXCR2 gene, we found significant dawn expression in all our patients. Several in vitro and in vivo studies revealed that CXCR2 antagonize resulted in a great cancer progression as well as metastasis inhibition; thus, it could be a promising strategy for cancer therapy [59,60].

Interleukin-13 (IL-13) is a cytokine that plays a significant role in negatively regulating anti-tumor immunity. It has been found to have a pivotal role in promoting tumor growth and progression by enabling cancer cells to evade the immune system [61]. However, we found evidence of a significant down-regulation in the IL13 gene. This finding is consistent with previous research in cancer therapy, which has suggested that blocking the IL13 gene can have potential benefits for patients undergoing therapy [62,63]. The current study showed that CSF3 mRNA levels were down-regulated in both HPV-associated and negative TSCC patients. Several studies have uncovered molecular pathways and immune responses in the colorectal TME, indicating that targeting CSF3 could be a potential strategy for cancer treatment [64].

The immune microenvironments of HPV-associated and HPV-non-associated OPSCC were distinct. HPV-associated OPSCC is characterized by the up-regulation of numerous immunogens and immune signaling pathways, as well as immune cell infiltration [65,66]. Our results showed that the HPV interaction might play a critical role in modulating the TSCC microenvironment, thus leading to poor clinical outcomes. Targeting those genes with antagonized molecules may be a novel strategy for TSCC treatment. Further differentiation of the immune-related genes of TSCC is necessary to categorize patients who may benefit from immunotherapy. CCL4, CXCL12, and CSF3 strongly increased gene expression in the TSCC patients with HPV compared to the TSCC patients without HPV. It is important to mention that two genes, namely IL-6 and CSF2, were not changed between the two groups. Regarding down-regulation, we found that the HPV-negative group was characterized by highly down-expressed immune-related genes (CXCR2, CXCL9, IL13, CSF3, and CCR7) compared to the HPV-positive patients. Our results showed that the HPV interaction might play a critical role in modulating the HNSCC microenvironment, thus leading to poor clinical outcomes. Targeting those genes with antagonized molecules may be a novel strategy for HNCC treatment.

Genetic stability is responsible for tumor initiation and progression. We aimed to study 84 genes related to oncogenesis promotion, tumor suppressor genes, transcription factors, angiogenesis, and apoptosis. Those genes were categorized into different clusters: the oncogene cluster; the tumor suppressor cluster; the transcription factors cluster; the angiogenesis cluster; the cell adhesion molecules cluster; and the cell cycle cluster. We found that 48 genes were different in their expression patterns. In the HPV-negative TSCC patients, only 11 genes were up-regulated. In head and neck cancer, several oncogene families have been shown to play an oncogenic role. It has been reported that oncogenes play an important role as prognostic and tumor-prognostic factors in many types of cancer [67]. During the last few years, targeted therapy in head and neck cancer that aims to manipulate over-expressed oncogenes has shown some degree of promise [68]. However, we found that the tumor suppressor gene clusters showed an over-expression of 14 genes. TSG has shown an important role in the regulation of the cell cycle and the control of cancer initiations [69]. An accumulation of evidence has shown that high-risk HPV plays a curtail role in the development of different types of cancers by inactivating tumor suppressor genes [70]. Transcription factors play a crucial role in cell functionality, and the deregulation of these factors can lead to carcinogenesis. HNSCCs were characterized by alterations in several transcription factors that were associated with increased metastasis, proliferation, and a decrease in survival rates [71]. In this study, we found that the HPV-non-associated TSCC patients had only 11 up-regulated genes, and only one gene (TSC1) was unique for this group. In cancer, PIK3C2A has been associated with the dysregulation of the phosphoinositide 3-kinase (PI3K)/AKT/mTOR pathway, which is frequently observed in various cancer types [72,73]. The abnormal activation of this pathway can contribute to the development and progression of cancer by promoting cell survival, growth, and invasion [72,73]. On the other hand, TSC1 is a tumor suppressor gene that is part of the TSC1–TSC2 complex [74,75]. This complex regulates the mammalian target of the rapamycin (mTOR) pathway, which plays a critical role in cell growth, proliferation, and metabolism [74,75,76]. These findings suggest that PIK3C2A and TSC1 may play a significant role in the development and progression of HPV-non-associated TSCC. However, further investigation is needed to determine the specific role of these genes in HPV-non-associated TSCC. 

In this study, many key transcription factor genes were over-expressed in patients with TSCC. Until now, there has been a gap in our understanding of the mechanisms and how transcription factor deregulation contributes to tumor development, progression, and metastasis, as well as treatment resistance and disease clinical outcomes. It is quite a surprising finding that in one patient, we observed a rather remarkable down-regulation of 168 genes. This could be indicative of various underlying conditions, such as a genetic mutations. However, further research is necessary to ascertain the root cause of this phenomenon. In any case, this discovery could have significant implications for the development of new treatments and therapies for a wide range of tumors and diseases that are related to gene expression and regulation. A complete and deeper molecular understanding of the exact role of those genes in HPV-associated and HPV-non-associated TSCC could lead to identifying a potential target for effective cancer treatment approaches.

Globally, the most prevalent hr-HPV genotype detected in HNSCC is HPV-16, followed by HPV-31, HPV-18, HPV-56, HP-V52, HPV-33, and HPV-35, as well as other hr-HPV genotypes [8]. Interestingly, we found that the most common types were HPV-16 and HPV-35, followed by HPV-18, HPV-39, and HPV-45. The presence of HPV in the TSCC patients varied from single to multiple infections. Surprisingly, we detected quadruple HPV infections in the TSCC patients. Several studies have shown that multiple HPV infections are characterized by higher viral loads and an increased risk of cervical lesions compared to single HPV infections [77,78,79,80,81]. However, previous studies have shown that single HPV infections have a greater risk of initiating TSCC compared to multiple infections [77,78]. Patients with HPV-positive OPSCC showed increased survival rates and were found to be more responsive to chemo/immunotherapy and radiation therapy than their HPV-negative counterparts [82]. Overall, it is crucial to prevent HPV infection through vaccination, screening, and treating both single and multiple HPV infections to reduce the risk of developing HPV-related cancers. In summary, recent research has shown that the tumor microenvironment (TME) of head and neck squamous cell carcinoma (HNSCC) is a complex and diverse environment. The TME is known to play a crucial role in cancer progression and has been identified as a key factor in determining responses to therapy. Therefore, understanding the heterogeneity of the TME in HNSCC is of utmost importance in developing effective therapeutic strategies for this disease.

The HPV genome codes for six oncogenes named E1, E2, E4, E5, E6, and E7 [12,13]. These genes play a crucial role during viral replication and tumorigenesis [13]. E6 degrades the tumor suppressor p53 and ensures the survival of infected cells, while E7 promotes proliferation and inactivates the retinoblastoma protein (pRB) [14]. E6 and E7 could also cause genomic instability. The integration of HPV DNA into the human genome and E6 and E7 persistent expression is considered a key event in the progression of HPV-associated cancers [15]. In the current study, we were able to detect higher expression levels and integration of HPV 16 E7 in TSCC patients. Several studies have examined the effect of HPV-related genes on tumor heterogeneity in different types of cancer, including cervical and oropharyngeal cancers [83,84]. Recently, several current therapies targeting HPV E6 and E7 using a single or a combination of E6/E7 inhibitors and immunotherapy have been effective in pre-clinical studies [85,86]. The exact consequence of HPV-related oncogenes on tumor heterogeneity is not completely understood. However, HPV oncogene integration promotes tumorigenesis, and intra-tumor genetic heterogeneity is associated with poor cancer therapy responses.

Limitations should be acknowledged for this research. Firstly, this study is focused on a specific cohort of TSCC patients from a single hospital, which may limit the generalizability of the findings. Secondly, we used a relatively small sample size, which may affect the statistical power and generalizability of the results. Further research should be conducted with larger samples. Thirdly, the study did not investigate the functional implications of the identified over-expressed genes in the HPV-associated and HPV-non-associated TSCC patients, which could provide further insights into the underlying mechanisms.

## 5. Conclusions

The TMEs of HPV-associated and HPV-non-associated TSCC have shown distinct and heterogeneous gene expression related to various cellular mediators and factors involved in inflammation, immunity crosstalk, transcription factors, immune signaling pathways, signal transduction, oncogenesis, tumor suppression, angiogenesis, and apoptosis. Further in vitro and in vivo studies are needed to explore the functional implications of the identified over-expressed genes in HPV-associated and non-associated TSCC patients, which could provide more insights into the underlying mechanisms. These experiments can provide crucial insights into the biological effects of these genes and their relevance in different subtypes of TSCC, identify potential therapeutic targets, and pave the way for personalized treatment strategies. Additionally, more in-depth molecular pathways and immunological studies are required to assess the potential of targeting genes for cancer therapy.

## Figures and Tables

**Figure 1 cancers-15-05548-f001:**
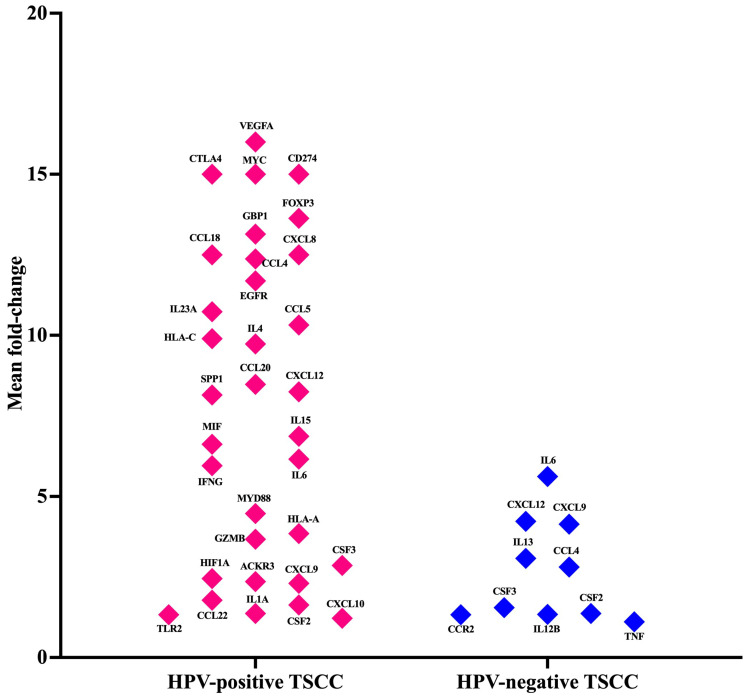
The mean fold-change gene expression is visualized for 37 genes related to communication between tumor cells and the cellular mediators of inflammation and immunity crosstalk in HPV-positive and HPV-negative TSCC patients. (Fold-change > 1.5).

**Figure 2 cancers-15-05548-f002:**
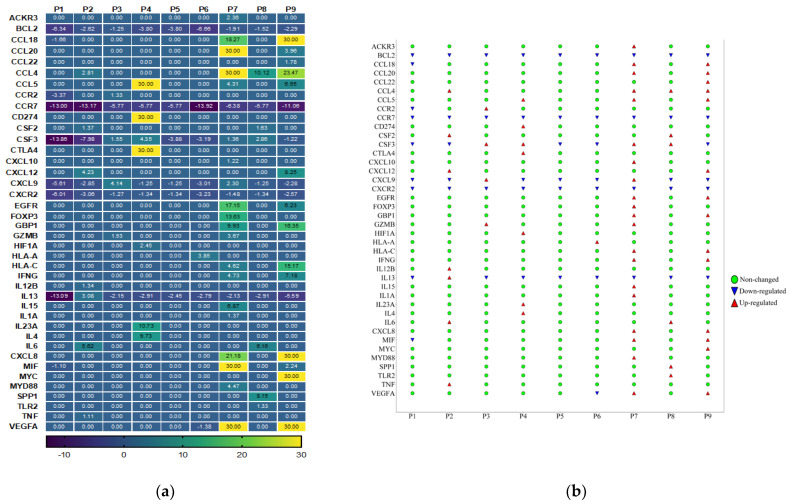
(**a**) Heatmap of expression profiles for the 40 dysregulated genes in HPV-associated and non-associated HPV TSCC patients. The yellow through to dark blue colors indicate high to low expression levels. The first left three columns indicate patients (P1, 2, and 3) with HPV-non-associated TSCC, and the right six columns indicate HPV-associated TSCC patients (p4–9). (**b**) Detailed dysregulated and unchanged gene for each patient (fold change > 1.5).

**Figure 3 cancers-15-05548-f003:**
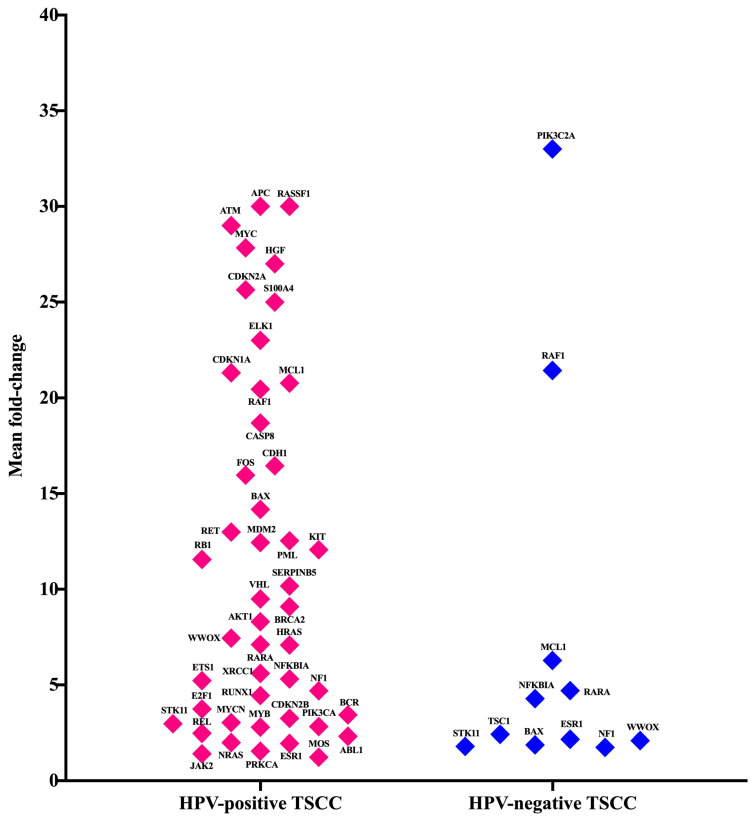
The mean fold-change gene expression is visualized for 48 genes related to oncogenesis and tumor suppression in HPV-positive and HPV-negative TSCC patients. (Fold-change > 1.5).

**Figure 4 cancers-15-05548-f004:**
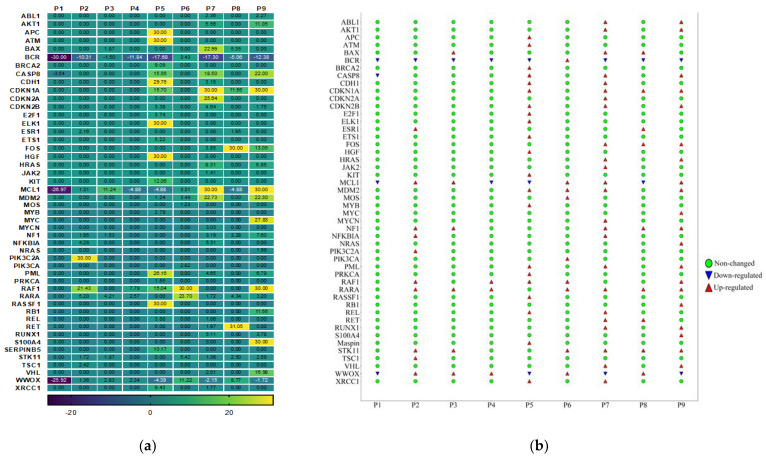
(**a**) Heatmap of expression profiles for the 48 dysregulated genes in HPV-associated and non-associated HPV TSCC patients. The yellow through to dark blue colors indicate high to low expression levels. The first left three columns indicated patients (P1, 2, and 3) with HPV-non-associated TSCC, and the right six columns are HPV-associated TSCC patients (p4–9). (**b**) Detailed dysregulated and unchanged gene for each patient (fold change > 1.5).

**Figure 5 cancers-15-05548-f005:**
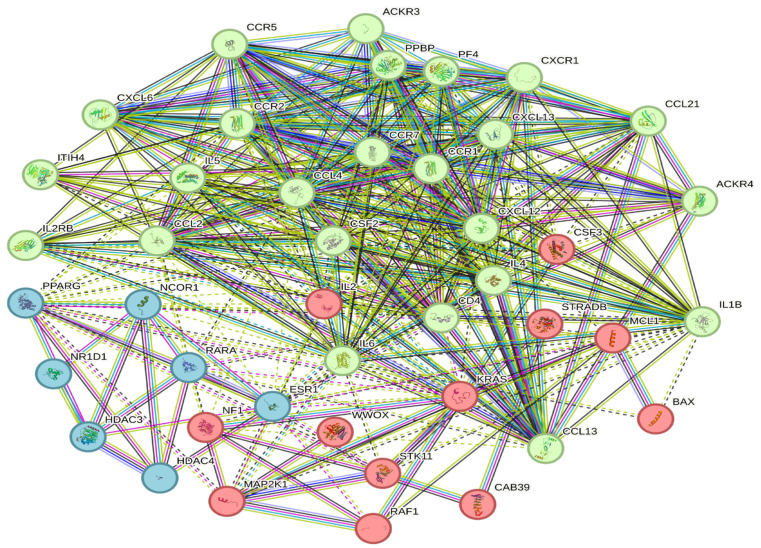
STRING PPI interactions between proteins are shown by lines with different colors. The edges represent protein–protein associations. Blue and purple edges indicate experimental evidence and interactions with known co-occurrence evidence, respectively. Yellow and black edges indicate co-expression and text mining, respectively. Different colors indicate various types of interactions.

**Figure 6 cancers-15-05548-f006:**
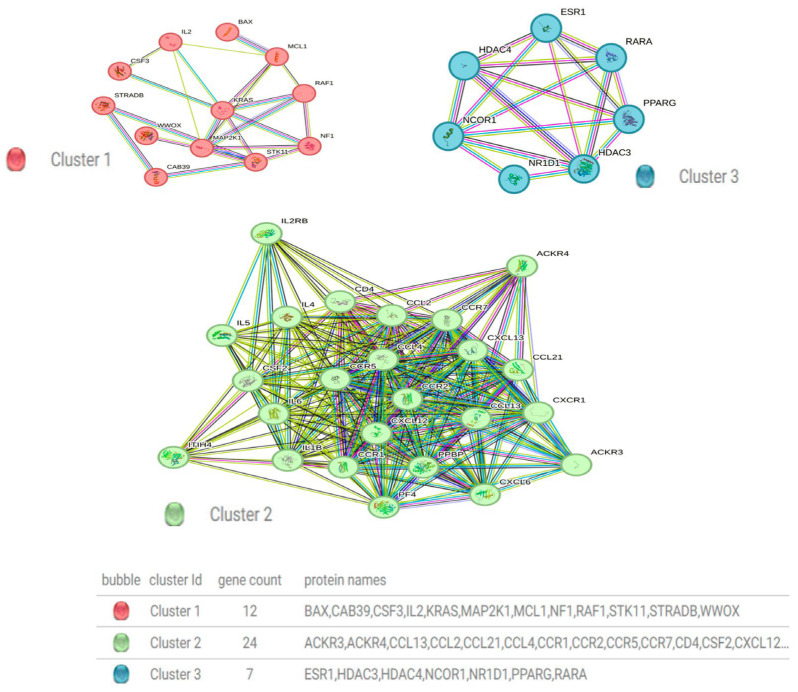
STRING PPI interactions between proteins are shown by lines with different colors. The edges represent protein–protein associations. Blue and purple edges indicate experimental evidence and interactions with known co-occurrence evidence, respectively. Yellow and black edges indicate co-expression and text mining, respectively. Different colors indicate various types of interactions.

**Table 1 cancers-15-05548-t001:** Demographics and clinical characteristics.

Baseline Variables	All Patients	HPV-Positive	HPV-Negative
Demographics	*n* = 16 (%)	*n* = 13 (%)	*n* = 3 (%)
Age (years)			
Median	57 ± 12	52 ± 13	59 ± 4
Range	24–70	24–70	58–67
Gender			
Male	7 (43.7%)	5 (38.5%)	2 (66.7%)
Female	9 (56.3%)	8 (61.5%)	1 (33.3%)
Ethnicity			
Saudi	13 (81.2%)	10 (53.8%)	3 (100%)
Non-Saudi	3 (18.8%)	3 (23%)	-
Smoking	4 (25%)	2 (15.3%)	2 (66.7%)
Comorbidities			
Diabetes mellitus	3 (18.8%)	2 (15.3%)	1 (33.3%)
Hypertension	4 (25%)	2 (15.3%)	2 (66.7%)

**Table 2 cancers-15-05548-t002:** Prevalence of HPV genotypes in (TSCC).

Variables	*n* (%)
**Type of HPV infection (*n* = 13)**	
Single	5 (38%)
Multiple	8 (61%)
HPV genotypes	
**Single infection (*n* = 5)**	
HPV16	3 (60%)
HPV35	2 (40%)
**Multiple infection (*n* = 8)**	
Double infections	2 (25%)
Triple infections	5 (62%)
Quadruple infection	1 (12%)

## Data Availability

The data presented in this study are available on request from the corresponding author.

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
