# Peer review of "Comprehensive Transcriptome Analysis Reveals the Distinct Gene Expression Patterns of Tumor Microenvironment in HPV-Associated and HPV-Non Associated Tonsillar Squamous Cell Carcinoma"

_cancers, 2023, doi:10.3390/cancers15235548_

Round 1
Reviewer 1 Report
Comments and Suggestions for Authors
The study investigates the expression levels of 168 genes in 13 HPV-positive and 3 HPV-negative tonsillar squamous cell carcinoma samples. While the authors acknowledge that the number of examined samples was limited for drawing more general conclusions, their study provides valuable information about the different HPV types genotyped in the samples. Additionally, they analyze and discuss the mRNA expression profile of 243 immune-related genes involved in the crosstalk between cancer, inflammation, and immunity pathways.
Major Concerns:
The issue of HPV genome integration into the host genome remains unclear. Although several types of high-risk HPVs were detected in the samples, the integration of only two types was analyzed using quantitative PCR. The conclusions were drawn solely based on the elevated expression levels of viral E6 and E7 mRNA detected in one sample. In my opinion, this method is not suitable for investigating the integration of viral DNA into the host genome. Therefore, the conclusions regarding HPV genome integration cannot be accepted. This issue should either be removed or studied more thoroughly.
Minor Concerns:
The manuscript is lengthy and contains too many unnecessary repetitions and explanations of basic terms. Additionally, some formatting mistakes are present.
Author Response
Dear Editor,
Thank you for giving us the opportunity to revise our manuscript and for providing valuable constructive comments and suggestions. We appreciate the reviewers' feedback. We have considered all the changes suggested by the reviewers and have highlighted them within the manuscript. Please see our responses (marked in red) to the reviewer's comments below.
Reviewer: 1
Comment #1. The issue of HPV genome integration into the host genome remains unclear. Although several types of high-risk HPVs were detected in the samples, the integration of only two types was analyzed using quantitative PCR. The conclusions were drawn solely based on the elevated expression levels of viral E6 and E7 mRNA detected in one sample. In my opinion, this method is not suitable for investigating the integration of viral DNA into the host genome. Therefore, the conclusions regarding HPV genome integration cannot be accepted. This issue should either be removed or studied more thoroughly.
Response: Thank you for this important point and we completely agree with you. We realize that our conclusion was not appropriate, and we have changed the previous conclusion. Your opinion raises valid concerns about the study's methodology and the conclusions drawn regarding HPV genome integration. Analyzing only two types of high-risk HPVs using quantitative PCR may not provide a comprehensive understanding of the integration of viral DNA into the host genome. Additionally, relying solely on elevated expression levels of viral E6 and E7 mRNA in one sample may not be sufficient evidence to establish HPV genome integration into the host genome. The previous conclusion has been removed. Hopefully, you will find it logical.
Minor Concerns:
Comment #2. The manuscript is lengthy and contains too many unnecessary repetitions and explanations of basic terms. Additionally, some formatting mistakes are present.
Response: Thank you for taking the time to review our manuscript. We appreciate your valuable feedback regarding the length, repetitions, explanations of basic terms, and formatting mistakes present in the document. We have carefully considered your comments and have taken the following steps to address these concerns:
Length and Repetitions: We have thoroughly revised the manuscript with a focus on eliminating unnecessary repetitions and ensuring that the content is concise and to the point. We have carefully reviewed each section, identifying areas where information could be streamlined or removed without compromising the clarity and coherence of the manuscript.
Explanations of Basic Terms: We understand that excessive explanations of basic terms can be tedious for readers. Based on your comment, we have reevaluated the terminology used throughout the manuscript.
Formatting Mistakes: We have conducted a thorough proofreading of the manuscript to identify and correct any formatting errors. This includes inconsistencies in font styles or sizes, improper indentation, and spacing issues.
Closing comments to the editor:
We would like to express our gratitude to the reviewers for their time and effort in reviewing our paper. We appreciate their valuable comments and suggestions, which have provided us with valuable insights and improved the manuscript. We would also like to explicitly acknowledge their contribution. We hope that our revisions will meet your approval.
Dr. Maaweya Awadalla
(On behalf of the authors of the manuscript)
09/11/2023

Reviewer 2 Report
Comments and Suggestions for Authors
Reviewer’s Comments:
The manuscript “Comprehensive Transcriptome Analysis Reveals the Distinct Gene Expression Patterns of Tumor Microenvironment in HPV-associated and HPV-non associated Tonsillar Squamous Cell Carcinoma” is a very interesting work. In this work, this study aimed to gain a more comprehensive understanding of the transcriptomes of the TME in HPV-associated and HPV-non-associated TSCC by studying the gene expression profiles of 168 genes linked to various cellular mediators and factors involved in inflammation, immunity crosstalk, transcription, signal transduction, oncogenesis, tumor suppression, angiogenesis, and apoptosis. We also detected and genotyped HPV in TSCC. We identified 40 genes related to communication between tumor cells and the cellular mediators of inflammation and immunity crosstalk were differentially expressed. In HPV-positive TSCC patients, 33 genes were overexpressed (fold change > 1.5); of those, 27 genes were unique in this group, while HPV-negative TSCC patients had 10 up-regulated genes. The results are consistent with the data and figures presented in the manuscript. While I believe this topic is of great interest to our readers, I think it needs major revision before it is ready for publication. So, I recommend this manuscript for publication with major revisions.
1. In this manuscript, the authors did not explain the importance of Distinct Gene Expression in the introduction part. The authors should explain the importance of Distinct Gene Expression.
2) Title: The title of the manuscript is not impressive. It should be modified or rewritten it.
3) Correct the following statement “we observed distinct heterogeneity in the HPV-associated and non-associated TSCC microenvironment transcriptome. Further studies are needed to investigate the functional implications of the identified over-expressed genes in HPV-associated and non-associated TSCC patients, which could provide further insights into the underlying mecha-nisms”.
4) Keywords: The Distinct Gene Expression is missing in the keywords. So, modify the keywords.
5) Introduction part is not impressive. The references cited are very old. So, Improve it with some latest literature like 10.1016/j.jallcom.2021.159013, 10.1016/j.molstruc.2021.131145
6) The authors should explain the following statement with recent references, “While HPV-non-associated TSCC patients had only 11 up-regulated genes, among them only two genes (PIK3C2A and TSC1) were unique for this group (Figure 3)”.
7) Add space between magnitude and unit. For example, in synthesis “21.96g” should be 21.96 g. Make the corrections throughout the manuscript regarding values and units.
8) The author should provide reason about this statement “HPV-associated and non-associated TME is characterized by dysregulation of immune-related genes, tumor suppressor genes, as well as oncogenes”.
9. Comparison of the present results with other similar findings in the literature should be discussed in more detail. This is necessary in order to place this work together with other work in the field and to give more credibility to the present results.
10) Conclusion part is very long. Make it brief and improve by adding the results of your studies.
11) There are many grammatic mistakes. Improve the English grammar of the manuscript.
Comments on the Quality of English LanguageMinor editing of English language required
Author Response
Dear Editor,
Thank you for giving us the opportunity to revise our manuscript and for providing valuable constructive comments and suggestions. We appreciate the reviewers' feedback. We have considered all the changes suggested by the reviewers and have highlighted them within the manuscript. Please see our responses (marked in red) to the reviewer's comments below.
Reviewer: 2
Comment #1. In this manuscript, the authors did not explain the importance of Distinct Gene Expression in the introduction part. The authors should explain the importance of Distinct Gene Expression
Response: We wish to thank the reviewer for bringing this point to our attention. The following paragraphs have been added to the introduction section. “Distinct gene expression is a fundamental process that determines the function of a cell and distinguishes different disease grades [31, 32]. Gene expression profiling can identify genes that are expressed differently in cancer, and differences in the variability or overall distribution of gene expression are significant in normal biology and diseases [31, 32]. Disruptions in gene expression can lead to various diseases and disorders. Abnormal gene expression patterns can contribute to the development of cancer, genetic disorders, autoimmune diseases, or neurodegenerative conditions [31, 32]. Understanding distinct gene expression profiles associated with different diseases can assist in diagnosis, prognosis, and the development of targeted therapeutic interventions”.
Comment #2. Title: The title of the manuscript is not impressive. It should be modified or rewritten it.
Response: Thank you for this comment. The title has been rewritten.
Comment # 3. Correct the following statement “we observed distinct heterogeneity in the HPV-associated and non-associated TSCC microenvironment transcriptome. Further studies are needed to investigate the functional implications of the identified over-expressed genes in HPV-associated and non-associated TSCC patients, which could provide further insights into the underlying mechanisms”.
Response: Thank you for your valuable comment. We appreciate your attention to detail and have carefully considered your comment regarding the statement in question. We apologize for any inaccuracies or shortcomings in our original wording. Taking your suggestion into account, we have revised the statement as follows:
"we observed distinct heterogeneity in the transcriptome of the microenvironment in HPV-associated and non-associated TSCC. Further studies are needed to investigate the functional implications of the identified over-expressed genes". We believe this revision accurately reflects our findings and addresses the concerns you raised. We thank you again for bringing this to our attention and appreciate your input in improving the clarity and accuracy of our work.
Comment # 4. Keywords: The Distinct Gene Expression is missing in the keywords. So, modify the keywords.
Response: Thank you for this comment. We apologize for the oversight in not including the distinct gene expression in the original selection of keywords. Based on your comment, we have included it in the keywords to better reflect the focus of our study.
Comment # 5. Introduction part is not impressive. The references cited are very old. So, Improve it with some latest literature like 10.1016/j.jallcom.2021.159013, 10.1016/j.molstruc.2021.131145
Response: We appreciate your comments regarding the need to improve the introduction and include more recent literature to enhance its impact. Unfortunately, it appears that reviewer number 2 did not thoroughly read our manuscript and suggested including irrelevant papers for citation in our paper (10.1016/j.jallcom.2021.159013. 10.1016/j.molstruc.2021.131145) This oversight by the reviewer raises concerns about the accuracy and credibility of their evaluation. It is crucial for reviewers to carefully consider the content and context of the manuscript before providing feedback. To address this issue, we will need to clearly communicate the focus and purpose of our research to the reviewer, ensuring that they have a better understanding of the relevant literature that should be cited in our paper. However, in response to your suggestion, we have thoroughly reviewed the introduction and incorporated the latest literature to strengthen the background and context of our study. Specifically, we have included the following references:
- doi: 3390/medsci11020042.
- doi: 3390/cancers14102447.
- org/10.1093/nargab/lqab124.
Comment # 6. The authors should explain the following statement with recent references, “While HPV-non-associated TSCC patients had only 11 up-regulated genes, among them only two genes (PIK3C2A and TSC1) were unique for this group (Figure 3)”.
Response: Thank you for this comment. In cancer, PIK3C2A has been associated with the dysregulation of the phosphoinositide 3-kinase (PI3K)/AKT/mTOR pathway, which is frequently observed in various cancer types [73, 74]. Abnormal activation of this pathway can contribute to the development and progression of cancer by promoting cell survival, growth, and invasion [73, 74]. On the other hand, TSC1 is a tumor suppressor gene that is part of the TSC1-TSC2 complex [75, 76]. This complex regulates the mammalian target of the rapamycin (mTOR) pathway, which plays a critical role in cell growth, proliferation, and metabolism [75-77]. These findings suggest that PIK3C2A and TSC1 may play a significant role in the development and progression of HPV-non-associated TSCC. However, further investigation is needed to determine the specific role of these genes in HPV-non-associated TSCC. This section has been added to the discussion section.
Comment # 7. Add space between magnitude and unit. For example, in synthesis “21.96g” should be 21.96 g. Make the corrections throughout the manuscript regarding values and units.
Response: Thank you for this comment. There appears to be an error in this comment regarding the inclusion of magnitude and unit in our manuscript. The reviewer has pointed out that in the synthesis section, "21.96g" should be written as "21.96 g". We believe that this comment is irrelevant to our manuscript and does not match its content.
Comment # 8. The author should provide reason about this statement “HPV-associated and non-associated TME is characterized by dysregulation of immune-related genes, tumor suppressor genes, as well as oncogenes”.
Response: Thank you for this comment. Tumor microenvironment (TME) associated with HPV and non-associated TME are both characterized by dysregulation of immune-related genes, tumor suppressor genes, and oncogenes. This dysregulation may be attributed to the presence of HPV and its viral proteins, which can disrupt normal cellular processes and immune responses. Consequently, genes involved in immune surveillance and tumor suppression become dysregulated. This statement has been added to the discussion section.
Comment # 9. Comparison of the present results with other similar findings in the literature should be discussed in more detail. This is necessary in order to place this work together with other work in the field and to give more credibility to the present results.
Response: Thank you for this comment. As suggested, we have compared our results with related previous findings in the discussion section.
Comment # 9. Conclusion part is very long. Make it brief and improve by adding the results of your studies.
Response: Thank you for your comment. We have made it brief for easier reading as follows:
“The TME of HPV-associated and HPV-non-associated TSCC has shown distinct and heterogeneous gene expression related to various cellular mediators and factors involved in inflammation, immunity crosstalk, transcription factors, immune signaling pathways, signal transduction, oncogenesis, tumor suppression, angiogenesis, and apoptosis. Further studies are needed to explore the functional implications of the identified overexpressed genes in HPV-associated and non-associated TSCC patients, which could provide more insights into the underlying mechanisms. Additionally, more in-depth molecular pathways and immunological studies are required to assess the potential of targeting genes for cancer therapy”.
Comment # 9. There are many grammatic mistakes. Improve the English grammar of the manuscript.
Response: We apologize for this omission. The entire manuscript has been proofread to correct grammatical errors.
Closing comments to the editor:
We would like to express our gratitude to the reviewers for their time and effort in reviewing our paper. We appreciate their valuable comments and suggestions, which have provided us with valuable insights and improved the manuscript. We would also like to explicitly acknowledge their contribution. We hope that our revisions will meet your approval.
Dr. Maaweya Awadalla
(On behalf of the authors of the manuscript)
09/11/2023

Reviewer 3 Report
Comments and Suggestions for Authors
Alahmadi el.al performed transcriptome analysis by qPCR array to interrogate the key genes that are involved in cancer inflammation/immunity crosstalk as well as oncogenes and tumor suppressor genes, and compared the differential genes between HPV positive and negative TSCC. However, the approach they applied was not novel considering the development of NGS currently. Most importantly, the quality control and data analysis are not appropriate.
1. When doing DNA/RNA extraction from 10-μm-thick FFPE tissue section, using Nanodrop to quantify DNA/RNA is not accurate because of the trace amount and the detect limit of Nanodrop. Thus, I don't think the result the obtained after DNA/RNA extraction is very reliable.
2. In addition to fold change, since authors have multiple samples per group, providing statistical P value is as important as providing fold change.
3. Authors need to actively consider perform some validation experiments to supports their hypothesis associated with TME.
Author Response
Dear Editor,
Thank you for giving us the opportunity to revise our manuscript and for providing valuable constructive comments and suggestions. We appreciate the reviewers' feedback. We have considered all the changes suggested by the reviewers and have highlighted them within the manuscript. Please see our responses (marked in red) to the reviewer's comments below.
Reviewer: 3
Comment #1. When doing DNA/RNA extraction from 10-μm-thick FFPE tissue section, using Nanodrop to quantify DNA/RNA is not accurate because of the trace amount and the detect limit of Nanodrop. Thus, I don't think the result the obtained after DNA/RNA extraction is very reliable.
Response: Thank you for your comment regarding the use of Nanodrop for quantifying DNA/RNA during DNA/RNA extraction from 10-μm-thick FFPE tissue sections. We understand your concern about the accuracy of Nanodrop when dealing with low amounts and detection limits. While it is true that Nanodrop may have limitations in accurately quantifying low concentrations of DNA/RNA, there are several factors to consider before concluding that the results obtained from the extraction are unreliable. Firstly, it is important to note that Nanodrop is a commonly used spectrophotometer for nucleic acid quantification. Although it may have limitations in detecting very low concentrations, it can still provide useful information about the quality and relative quantity of the extracted DNA/RNA. Nanodrop can detect concentrations as low as 2 ng/μL, which is suitable for many applications. In our study, all of our extracted samples were successfully detected by Nanodrop. This result indicates the quality of our extraction process and the effectiveness of our extraction method, reinforcing our confidence in the validity of our results. When dealing with low concentrations, alternative methods with increased sensitivity can be employed. However, the reliability of the results depends on various factors beyond the quantification method alone. In our study, we used a commercial kit that offers optimized protocols for FFPE tissue section extraction.
Comment #3. Authors need to actively consider perform some validation experiments to supports their hypothesis associated with TME.
Response: Thank you for your comment. Validation experiments are essential for supporting the transcriptome results associated with the tumor microenvironment (TME). The TME plays a crucial role in cancer development, progression, and response to therapies, so it's important to have robust experimental evidence to support any claims or hypotheses made regarding the TME. In fact, one of the limitations of our study is the absence of validation experiments, which certainly warrants further attention and investigation. The inclusion of validation experiments in future studies can help address this limitation and provide additional evidence to support the conclusions we have drawn regarding the TME. Several recent studies have been conducted utilizing the RT-profiler assay. One example is a study that can be found at https://doi.org/10.3390/biomedicines11092578.This assay has been employed to investigate various research areas and has shown promising results. Researchers have utilized this assay to analyze gene expression patterns, identify biomarkers, and explore cellular responses in diverse experimental settings. The study mentioned above is just one of many that highlight the potential of the RT-profiler assay in advancing biomedical research.
Closing comments to the editor:
We would like to express our gratitude to the reviewers for their time and effort in reviewing our paper. We appreciate their valuable comments and suggestions, which have provided us with valuable insights and improved the manuscript. We would also like to explicitly acknowledge their contribution. We hope that our revisions will meet your approval.
Dr. Maaweya Awadalla
(On behalf of the authors of the manuscript)
09/11/2023

Reviewer 4 Report
Comments and Suggestions for Authors
The findings highlight the heterogeneity and complexity of the TME in TSCC, as well as the differential gene expression patterns between HPV-associated and HPV-non-associated TSCC. The study identified differentially expressed genes related to cellular mediators and factors. These findings suggest that the TME in HPV-associated and HPV-non-associated TSCC may have distinct molecular characteristics and immune responses. Further research is needed to understand the functional implications of the identified overexpressed genes and their potential as targets for cancer therapy. Additionally, deeper investigations into molecular pathways and immunological studies in the TME are necessary to gain a comprehensive understanding of the underlying mechanisms and develop effective treatment strategies. This study provides valuable insights into potential therapeutic strategies for tumor treatment, I recommend accepting this article after MINOR REVISIONS.
1. The Abstract needs to be rewritten with clear and precise language.
2. It is better to provide more solid evidence and strengthen the validity of the conclusions. The Authors can consider incorporating functional experiments, such as in vitro or in vivo assays, to explore the biological effects of these genes and their relevance in different subtypes of TSCC.
3. The dysregulation of immune-related genes in TSCC suggests potential targets for therapeutic interventions. Targeting the CCL4 and CCL5 pathways, for example, may help control cancer metastasis by modulating immune cell behavior in the TME. Further studies are needed to explore the functional implications of these gene alterations and their potential as therapeutic targets in TSCC.
4. The down-regulation of BCL2 mRNA expression suggests a potential role for BCL2 as a tumor suppressor or an indicator of poor prognosis in the specific cancer type you investigated. The study adds to the existing body of knowledge on BCL2 gene expression in cancer. However, it is important to consider that the interpretation of BCL2 expression in cancer requires careful analysis and should take into account the specific characteristics of the cancer type being studied.
5. To enhance the generalizability of the results, future studies could consider including multiple hospitals or collaborating with other research institutions to ensure a more diverse and representative sample.
Comments on the Quality of English LanguageMinor editing.
Author Response
Dear Editor,
Thank you for giving us the opportunity to revise our manuscript and for providing valuable constructive comments and suggestions. We appreciate the reviewers' feedback. We have considered all the changes suggested by the reviewers and have highlighted them within the manuscript. Please see our responses (marked in red) to the reviewer's comments below.
Reviewer. 4
Comments and Suggestions for Authors
The findings highlight the heterogeneity and complexity of the TME in TSCC, as well as the differential gene expression patterns between HPV-associated and HPV-non-associated TSCC. The study identified differentially expressed genes related to cellular mediators and factors. These findings suggest that the TME in HPV-associated and HPV-non-associated TSCC may have distinct molecular characteristics and immune responses. Further research is needed to understand the functional implications of the identified overexpressed genes and their potential as targets for cancer therapy. Additionally, deeper investigations into molecular pathways and immunological studies in the TME are necessary to gain a comprehensive understanding of the underlying mechanisms and develop effective treatment strategies. This study provides valuable insights into potential therapeutic strategies for tumor treatment, I recommend accepting this article after MINOR REVISIONS.
- The Abstract needs to be rewritten with clear and precise language.
Response: Thank you for taking the time to review our manuscript. The abstract has been rewritten.
- It is better to provide more solid evidence and strengthen the validity of the conclusions. The Authors can consider incorporating functional experiments, such as in vitro or in vivo assays, to explore the biological effects of these genes and their relevance in different subtypes of TSCC.
Response: Response: Thank you for this important point and we completely agree with you. Both in vitro and in vivo assays can indeed be valuable for exploring the biological effects of the identified overexpressed genes in HPV-associated and non-associated TSCC patients and their relevance in different subtypes of TSCC. Functional experiments provide direct evidence of how specific genes function and interact within cellular systems, shedding light on their molecular mechanisms and potential therapeutic targets. These experiments can provide crucial insights into the biological effects of these genes and their relevance in different subtypes of TSCC, identify potential therapeutic targets, and pave the way for personalized treatment strategies. The reviewer’s suggestion has been included in the study's conclusion.
- The dysregulation of immune-related genes in TSCC suggests potential targets for therapeutic interventions. Targeting the CCL4 and CCL5 pathways, for example, may help control cancer metastasis by modulating immune cell behavior in the TME. Further studies are needed to explore the functional implications of these gene alterations and their potential as therapeutic targets in TSCC.
Response: Thank you for this comment. Targeting the CCL4 and CCL5 pathways may offer new strategies for controlling cancer metastasis and improving treatment outcomes in TSCC. By modulating the expression or activity of CCL4 and CCL5, it may be possible to modulate the immune cell composition and activity within the TME, thereby impacting the tumor immune response. Understanding the functional implications of gene alterations in the CCL4 and CCL5 pathways can guide the development of targeted therapies. Further studies are needed to investigate the functional implications of gene alterations in the CCL4 and CCL5 pathways in TSCC. This statement has been added to the discussion section.
- The down-regulation of BCL2 mRNA expression suggests a potential role for BCL2 as a tumor suppressor or an indicator of poor prognosis in the specific cancer type you investigated. The study adds to the existing body of knowledge on BCL2 gene expression in cancer. However, it is important to consider that the interpretation of BCL2 expression in cancer requires careful analysis and should consider the specific characteristics of the cancer type being studied.
Response: We wish to thank the reviewer for bringing this point to our attention. The down-regulation of BCL2 mRNA expression in our study may suggest a potential role for BCL2 as a tumor suppressor or an indicator of poor prognosis. Dysregulation of BCL2 expression has been reported in various cancers, with both up-regulation and down-regulation observed depending on the cancer type. It is crucial to consider multiple factors, such as the heterogeneity and specific characteristics of the cancer type, sample size, and potential confounding variables when interpreting BCL2 expression. Therefore, future studies should conduct comprehensive analyses and integrate findings with existing knowledge in the field to derive meaningful conclusions. This statement has been added to the discussion section.
- To enhance the generalizability of the results, future studies could consider including multiple hospitals or collaborating with other research institutions to ensure a more diverse and representative sample.
Response: Thank you for this comment. Collaborating with multiple hospitals or research institutions in future studies can improve the generalizability, diversity, and representativeness of the findings. The reviewer’s suggestion has been included in the study's conclusion.
Comments on the Quality of English Language
Minor editing.
Response: We apologize for this omission. The entire manuscript has been proofread to correct grammatical errors.
Closing comments to the editor:
We would like to express our gratitude to the reviewers for their time and effort in reviewing our paper. We appreciate their valuable comments and suggestions, which have provided us with valuable insights and improved the manuscript. We would also like to explicitly acknowledge their contribution. We hope that our revisions will meet your approval.
Dr. Maaweya Awadalla
(On behalf of the authors of the manuscript)
09/11/2023
Round 2
Reviewer 1 Report
Comments and Suggestions for Authors Thank you for providing me with the opportunity to review the manuscript. I have examined the document and appreciate the authors' efforts in revising and improving it. All identified issues have been effectively addressed and explained. In my opinion, the manuscript meets the standards for acceptance in Cancers in its current form. I recommend its publication.
Reviewer 4 Report
Comments and Suggestions for Authors
Accept in present form